# Content Validation of a Semi-Structured Interview to Analyze the Management of Suffering

**DOI:** 10.3390/ijerph182111393

**Published:** 2021-10-29

**Authors:** Carmen Sánchez-Guardiola Paredes, Eva María Aguaded Ramírez, Clemente Rodríguez-Sabiote

**Affiliations:** Department of Methodology in Research and Diagnostic in Education, Universidad de Granada, 18071 Granada, Spain; eaguaded@ugr.es (E.M.A.R.); clerosa@ugr.es (C.R.-S.)

**Keywords:** expert judgement, content validity, suffering, concordance

## Abstract

This work involves the content validation of a semi-structured interview, whose objective is to learn about the management of suffering in people. The interview items have been classified into several categories that define the suffering construct. For the content validation of the instrument, in addition to initially conducting a scientific review on the subject, the procedure known as expert judgement has been used. The results obtained in terms of the content validity achieved in the dimensions and areas assessed are, in general, satisfactory. However, some of these dimensions and certain areas have not exceeded the required minimum values for content validity. Therefore, it is necessary to modify the items comprising these dimensions in the areas evaluated with the additional incorporation of the qualitative suggestions for improvement indicated by the experts. As for agreement among experts, the results point to moderate agreement, which, moreover, is not due to chance.

## 1. Introduction

The aim of many areas of knowledge, in particular, Philosophy, Psychology, Psychiatry, Medicine, Nursing, Sociology, Educational Sciences, Linguistics, and other related fields, has been for years to search for the most effective way to alleviate and manage human suffering in any of its facets. To this end, these various fields of knowledge have been explored in the scientific literature.

Human suffering and its management is a progressive topic of interest that affects human beings throughout their lives. Reference [1] states that suffering is individual, unique, and inherent to each person. It is characterized by its complexity and multidimensionality, where psychological, spiritual, socio-cultural, and familial needs, etc. must be addressed. A thorough understanding of the nature of suffering and its associated factors is required in order to alleviate unnecessary suffering.

All of this is interdisciplinary work as the human being is multidimensional. Reference [2] argues that it is not the body that suffers but the person, and the person is a unit, not a mind on one side and a body on the other. Moreover, Reference [3] agrees along the same lines:

We are complex units where the objective and the subjective, the environment and the biography are integrated, but we are units, therefore, those who suffer are us as a whole, not just the mind or the mind on the one hand and the body on the other. No. It is the whole, the unity (p. 74).

There are no fixed sources of human suffering. The impact of events is dependent on subjectivity. This idea has been defended by authors such as in [4,5,6,7], among others, who define suffering as a person’s attitude towards problems. Suffering is not only a lived reality, but also a certain way of interpreting that experienced reality. The interpretation of experience is not only an attempt to give sense or meaning to the phenomenon, but is one of its inherent elements. Each conceptualization makes sense on the basis of a series of premises implicit to that concept. In other words, on the basis of a set of previous interpretations that determine it. Behind each of the conceptualizations there is a narrative (about the world, human beings, good and evil, and other matters) that drives the concept of suffering. Therefore, every conceptualization of suffering presupposes an interpretation derived from a previous, implicit, and determinant, although invisible, narrative.

Reference [8] argues that the treatment for suffering should be delivered by holistic programs that treat the whole person, given that people are multidimensional beings. For instance, the authors such as in [1,9,10,11] and others agree with this concept. In research such as in [11], it is argued that there is empirical evidence that an alternative pedagogical treatment of suffering by practitioners would be for people to be cared for via holistic programs, with treatments that take into account the various dimensions of the human being: The affective, cognitive, and emotional, in order to improve their quality of life. Alleviating suffering caused by illnesses through biomedical sciences is no longer sufficient. Moreover, the new framework encompasses both the personal and institutional dimensions. This perspective would be very enriching and helpful in the alleviation of suffering. Empirical data and evidence from this research show that with treatments that care for the human being holistically, levels of anxiety, depression and, ultimately, suffering are reduced in the majority of cases.

Opportunities exist for education professionals, social educators, and pedagogues to improve situations of suffering in society. The study of the topic of suffering can provide invaluable information for all of the researchers to be able to understand the nature of suffering in each life story. Studies on suffering by [7,12,13,14,15,16,17,18], among others, show that it is still a work in progress from a pedagogical point of view, but not from the psychological, medical or palliative care perspectives, etc. where studies are more advanced.

The authors such as in [19] stress the importance of educating people on how to manage suffering, as it leads to a high culture with the right conditions for the human to grow and mature. Notions in favor of compassion, fear, eliminating suffering, etc. are characterized as an effective and real denial of life. This type of education would produce beneficial effects. The educational perspective on the management of suffering presupposes preparing the human being according to the actual needs of everyday life. Reference [20] explains that pain has its usefulness in preventing or helping to expel present or potentially bad aspects of life, sometimes serving to protect the injured organ. A similar response can be given with regards to psychological and moral suffering. One of its essential functions is to teach, not in an abstract informative way but by its very vital negativity, just as pleasure, if properly embraced, has a positive function in life.

Reference [18] attempts to answer the question: What is the purpose of negative affective experience? Attempts have been made to advance the virtue-theoretic perspective on the value of suffering. The general view is that suffering is necessary for the cultivation and expression of important forms of virtue, without which a happy and flourishing life is impossible. Certain forms of suffering in appropriate circumstances are: (1) Virtuous motives. (2) The promotion of the development of virtues associated with strength of character, vulnerability, morality, and wisdom. (3) The communication of virtue to others, fostering the social virtues associated with justice, love, and trust.

Moreover, suffering is a very broad and very complex phenomenon, a concept which is difficult to delimit and understand in just one term. An attempt has been made to narrow down the concept of suffering by investigating four different issues in the groups of people interviewed.

Suffering due to a partner relationship;Suffering due to illness;Suffering due to a failure to adapt to the environment;Suffering due to financial problems.

As already classified by some authors, such as the great philosopher in [21], other factors that can give rise to suffering and frustration in the person are specified. These include sexuality and eroticism, fear of death, poor health, economic instability, unrecognized achievements, religion, and politics.

This type of emotional pain or suffering includes pain caused by the break-up of a partner relationship, pain due to the serious or chronic illness or the death of a loved one, pain due to a failure to adapt to the environment caused, for example, by estrangement from loved ones such as children, partners, and parents, etc., pain due to a major disappointment in any area, for example, financial and economic matters, etc. [10].

Therefore, the intention is to establish communication links between reality, theory, and practice in order to draw conclusions that improve the management of the suffering of these people, their families, and friends without ignoring improvements in emotional, educational, medical, and clinical management and care. In general, this is a multidisciplinary and holistic approach due to the multidimensional nature of human beings. In the field of social sciences, researchers typically design different types of questionnaires such as tests, surveys, interviews (structured, semi-structured, and unstructured), and scales, etc. to collect data that respond to their study variables. Sometimes standardized tests are used and other times the researcher develops a data collection instrument according to the needs and objectives of the particular study. These instruments consist of items that contain indicators formed by a theoretical framework that aims to make the constructs to be measured observable.

According to [22] (p. 17) “indicators tend to play a mediating role between the scientific literature and the empirical, i.e., between the theoretical framework and the external world”. Indicators point to and refer to observable, measurable, and empirically detectable features and characteristics.

The use of indicators for the efficient measurement or observation of a theoretical construct demands that these explanatory indicators satisfy a series of requirements. They must describe the reality that is going to be studied. Therefore, they must fulfil their function with a validity and reliability index in order for the results obtained from the research to have scientific rigor.

Validity in general has been argued by [23] (p. 1) as “the accuracy with which adequate and meaningful measurements can be made with a test”. In his opinion, some of the characteristics to be highlighted by measurement would be addressing increasing validity, basic values, accuracy in decision-making, homogeneity, and sequential development of the items. Similarly, the authors in [24] point out that content validity determines the degree to which an instrument reflects the specific content domain of what is to be measured. Moreover, it is responsible for assessing whether the instrument contemplates the dimensions of the construct to be measured. Therefore, the instrument is considered valid if it contemplates each and every one of the elements related to the concept of the construct [25,26].

Other authors such as in [27] state that content validity aims to guarantee that the indicators selected adequately represent the construct of interest, on the basis that the construction of the items is developed from the conceptualization of the variable to be measured.

The purpose of content validity is to provide evidence that the semantic definition is included in the items constructed, that they are relevant to the construct, and that they adequately address each of the dimensions proposed in the semantic definition [28].

Reference [17] explains that validity is usually studied using several components [17]: Content validity, criterion validity, and construct validity. The three are distinct and the use of each depends on the type of test being conducted. Researchers are generally interested in determining the content validity of measurement instruments such as self-developed questionnaires, as there is not usually any specific external criterion for these instruments.

Essentially, the so-called “expert judgement” procedure is used as a way of assessing content validity, in addition to expert agreement and the precise determination of whether the instrument is valid or not, and whether it actually measures what is required. Therefore, it is very important to quantify the degree of content validity of the instrument using indicators.

In this particular case, the validity was assessed in relation to all of the four aspects considered (content, wording, number, and relevance of the items). This procedure is useful as it improves the understanding of what is being measured in the test. If the test is valid, it is reliable, but not vice versa. From this, the content validity of the measurement instrument is determined by means of Aiken’s V coefficient and the interval scale, in order for the researchers to have guidance on handling the index and using it to further support the assessment made by the experts’ judgement.

Following this conceptual contribution, this work aims to validate an instrument to analyze the extent to which the items in the various categories are successful. The objective of this validation is to obtain an effective, reliable, and user-friendly instrument. The importance and need for the validation of data collection instruments have been the priority of many contemporary research studies, as References [29,30], among others, state in their work, and for whom the validation of the instrument is a crucial phase, which verifies that the data and conclusions obtained and practices carried out have or do not have sufficient substance and foundation.

## 2. Methods

### 2.1. Instrument

The instrument used in this research was the semi-structured interview specifically designed on an ad hoc basis and validated by expert judgement. Based on the categories that emerged as a result of the literature review, the objectives of this research will be answered. This interview has been called “the management of suffering in people suffering from illness, relationships, economic problems, and adaptation to the environment” (see Appendix A).

The sequence of items is divided into the following categories of analysis:Socio-demographic data;Suffering;Love in the dimension of family, friends, and partner;Acceptance;Non-acceptance;Resignation;Spiritual dimension;Verbal and non-verbal communication;Pain;Fear;Transience;Gratitude;Compassion;Hope;Palliative care;Sadness;Resilience;Happiness/life satisfaction, well-being.

Through each of these categories, a series of items have been elaborated in order to obtain the information addressed.

The data collection took place in different places as each informant preferred, although always in quiet places that safeguarded confidentiality. The process was recorded on audio and later transcribed.

Finally, we state that each of the interviews carried out is developed in a different way, taking into account the context and the reporting subject. The interviews have been recorded on audio with the consent of each of the interviewed subjects in order to preserve the collection of informative data.

Therefore, this research is framed within the proposed ethical considerations: Informed consent, avoiding deception of research participants, respecting participants’ privacy, upholding accuracy of data and interpretation, and respect for the individual. Moreover, we state that, with respect to the confidentiality of the participants, any identifying data that could recognize them have been removed, thus preserving anonymity in each and every one of the interviews. In addition, with respect to the consent document, each and every one of them was informed of the purpose of the research and agreed to it.

The semi-structured interviews provided an abundance of data, which were refined and analyzed to arrive at a final result. This research study was marked by data saturation.

### 2.2. Participants

It should be mentioned that the questionnaire is aimed at a sample of 22 respondents. With regards to the section on suffering due to illness, it was carried out with chronically ill patients from hospitals in Granada (Spain). On the part of the hospital, in a direct and intentional selection sample, those subjects with the highest degree of loneliness were assigned to the research. Therefore, the type of sampling was incidental. The participating sample consisted of four subjects of different genders between 41 and 80 years of age, from different cities in Spain and America.

With regards to the other sections on suffering due to economic problems, relationships, and adaptation to the environment, the participant sample was made up of informants of different genders between 40 and 65 years of age (Table 1, identification sheet) who provided data explaining the phenomenon of suffering until the information was saturated. Therefore, the sampling is non-probabilistic and intentional. As [31] points out, the final number of the sample is obtained when the informants do not present any more answers to the explanation of the phenomenon, reaching saturation of the testimonies or information.

Similarly, the so-called “snowball” technique was also used in sampling. This is a non-probability sampling technique used by researchers to identify potential subjects in studies where subjects are difficult to find. What is relevant in this research work is the importance of the word of the people interviewed and the information provided by them, as it is thanks to them that the results of this work will be obtained. These ideas have been defended by [32] who supports the study of narrative, as it is the way in which human beings experience the world. This general notion carries over to the conception that education is the construction and reconstruction of personal and social stories. In addition, that teachers and learners are narrators and characters of their own and others’ stories. This concept can be applied in the interaction between patient/educator, interviewee/educator, learner/educator as mutual learning takes place through the knowledge of the informants’ life stories. Thanks to these life stories, it will be possible to intervene in how to teach how to manage, lessen or alleviate suffering. As [33] explained, healing is bilateral, there is mutual teaching and learning on both sides.

### 2.3. Content Validation

In order to validate the semi-structured interview’s content, the assessment made by various experts on different aspects of the interview was taken into account. Reference [31] proposed a series of criteria for the selection of experts, among which we highlight:(a)Experience in performing judgements and making decisions, based on evidence on expertise, e.g., degrees, research, publications, etc.;(b)Reputation in community;(c)Availability and willingness to participate;(d)Impartiality and inherent qualities such as self-confidence and adaptability.

In our case, for the selection of experts we have considered a mixed criterion. In other words, their availability and willingness to participate and experience in the topic under evaluation based on evidence on expertise, namely research and publications. In this sense, a list of 10 experts from the university teaching profession assessed the semi-structured interview. All of them are people with extensive knowledge and proven experience in the area of interest and are therefore qualified to answer the questions posed. They come from different educational backgrounds, approach the problem experimentally as opposed to theoretically, and are from different professional experiences. They are intended to help improve the quality of the analysis.

In terms of their distinctive characteristics, it should be highlighted that a total of four men and six women aged between 45–65, with an average age of 55, participated. They belonged to different departments in the University of Granada (Pedagogy: 3; Business Organization: 2; Translation and Interpretation: 1; Political Science and Administration: 1; Voice Pedagogy: 1; CAMD: 1; Philanthropy and Business: 1). There was a range of professional categories, namely 3 tenured university professors, 1 retired tenured university professor, 1 associate professor, 1 tenured special education professor, 1 sports doctor, 1 on a permanent doctoral contract, 1 entrepreneur and philanthropist, and 1 on a postdoctoral contract. All this can be seen in Table 2.

Therefore, the validation of the content of the unstructured questionnaire has taken into account the assessment made by various experts on different aspects of the questionnaire through the so-called “expert judgement”. In addition, another ad hoc questionnaire has been drawn up. The experts were asked to evaluate the semi-structured interview with the items, taking into account the following aspects (see Table 3, validation scale):-Clarity of content: The questions are clearly and precisely worded, which makes them easy to understand for people suffering for different reasons.-Clarity of wording: The wording and terminology used are appropriate for the target audience.-Grouping of questions: Correspondence between the content of the question and the category in which it is placed. Logical order of presentation of questions.-Relevance of the data provided: The questions are relevant and provide the necessary data to answer the objectives.-Number of questions: The number of questions, for each of the objectives, is adequate, in a way that the interview does not become too long, in order to avoid interviewees finding it tedious to answer all of the items.

This document provides a guide to validation, structured in two parts: -Expert’s identification data;-Validation scale. The table below (one is given as an example) refers to the different objectives to which the data collection instrument responds. In this way, the evaluator will be able to assess the questions on each of the 17 categories according to the proposed criteria. In addition, a space is reserved for them to make any recommendations they consider appropriate and to suggest alternative ways of formulating the questions they consider inadequate.

Expert’s identification data:Sex: Male ( ) Female ( ) Intergender ( ) Other ( )Years of experience at university:Department to which it belongsCurrent professional category:

( ) University Professor

( ) University holder

( ) Permanent Doctoral Candidate

( ) Associate Professor

( ) Contracted Assistant Doctor

( ) Contracted Assistant

Subsequently, some items of the interview were modified and some recommendations were added, resulting in the final script of the semi-structured interview.

Through this validation scale, the various experts assess a total of 17 different categories (see Table 4), which have emerged as a result of the bibliographic review of the scientific literature that addresses suffering from various fields such as philosophy, psychology, generality, educational sciences, biomedical sciences, etc., since suffering is a phenomenon that must be treated holistically. Through each of these categories, a series of items have been elaborated in order to obtain the information addressed. Similarly, Reference [34] states that there may be room for other items that can be included in the course of the interview in order to delve deeper into the relevant issues that arise.

The 153 questions or items that make up the questionnaire are grouped into these 17 categories. In addition, 10 more items that make up the sample characteristics are included as identification data. These data include attributes such as age, gender, place of residence, ethnicity, level of education, religious affiliation, etc. of future participants who will fill in the unstructured questionnaire, whose object of measurement focuses on trying to measure how people manage suffering due to illness, relationships, economic issues, and lack of adaptation to the environment.

Each of these 17 categories was quantitatively assessed using a Likert scale (from 1: Not suitable at all, through to 4: Very suitable). This quantitative assessment refers to various factors, namely the clarity of the content of the questions and their wording, the adequacy of the number of questions, as well as the relevance of the data provided by the questions.

Finally, three dimensions of a qualitative nature are also included in the assessment protocol in order for the experts to express how they would modify the questions, if they consider it appropriate, which questions they would add, and which questions they would eliminate. However, this eminently qualitative part is not the subject of analysis and discussion in this paper.

### 2.4. Procedure and Statistics

For the validation of the content of the unstructured questionnaire, the purpose of which is to measure how people manage their level of suffering due to illness, relationships, economic issues, and failure to adapt to the environment, two different but complementary strategies have been considered.

On the one hand, the content validity of each of the 17 dimensions of the questionnaire was calculated individually in relation to the four dimensions considered. For this purpose, Aiken’s V validity coefficient was calculated up to 68 times (17 × 4) [46,47]. This index is useful for evaluating the importance of each item with respect to the construct being assessed. Its main advantage over similar indices, for example, [48], “is that it takes into account not only the number of categories available to the experts, but also the number of participating experts” [49] (p. 11).

Notwithstanding the above, the interpretation of this index was somewhat imprecise, since it was not associated with any statistical probability (p), nor with any confidence interval. For this reason, Reference [50] proposed a new reformulation of Aiken’s index based on the following terms:$$V=\frac{\mu -l}{k}$$
where $\bar{X}$ is the average of the judges’ ratings; *l* is the lowest obtainable score possible (1, in this case); and *k* is the difference between the highest and lowest score on the rating scale completed by the various experts (ranging from 1 to 4 points, as shown above, therefore *k* = 3 in this particular case).

For a meaningful interpretation of the same, Reference [50] adopted the so-called score method, whose advantage lies in its precision, despite the fact that the distribution here is asymmetric in nature. Moreover, these authors consider the Aiken index as a ratio to establish the interval at a given confidence level [51]. The equations for lower (L) and upper (U) confidence intervals are as follows:$$L=\frac{2nkV+z^{2}-z\sqrt{4nkV\left(1-V\right)+z^{2}}}{2\left(nk+z^{2}\right)}$$
$$U=\frac{2nkV+z^{2}-z\sqrt{4nkV\left(1-V\right)+z^{2}}}{2\left(nk+z^{2}\right)}$$
where *L* is the lower bound and *U*, the upper bound; *n* is the number of judges; *k* is the difference between the highest and lowest score on the rating scale; *V* is the value of Aiken’s V; and *z* is the chosen standard distribution. Therefore, 90%, 95%, and 99% confidence corresponds to 1.65, 1.96, and 2.58, respectively.

For the interpretation of confidence intervals, it is recommended that the lower limit should have a value ≥0.7 [52], although it is known that the confidence interval amplitude is highly dependent on the increase in sample size [50].

Furthermore, the need to take into account a global index of each of the dimensions of the four dimensions as a whole, i.e., globally, was considered. For this purpose, the most precise coefficient for this type of condition has been calculated, i.e., the intraclass correlation coefficient.

## 3. Results

Once the opinions of the 10 experts had been collected, two data matrices were drawn up with all of the information. The first matrix with an .xls extension is used to calculate the content validity of the 17 dimensions of the questionnaire considered (suffering, love, acceptance, etc.) in relation to the four aspects considered (content, wording, number, and relevance of the items). The second matrix with an SPSS .sav extension is used to calculate the agreement between the different experts measured on a metric scale (interval scale), for each of the four aspects considered (content, wording, number, and relevance of the items).

Therefore, from the first matrix, which has the Excel .xls extension, the Aiken V content validity index for each of the 17 dimensions in four differential aspects was calculated (17 × 4 = 68) (see Table 5). For this purpose, the application of [53] was used. The results obtained in this respect are shown below.

As can be seen, the content validity values obtained in the 68 dimensions assessed are, in general, satisfactory (they are worthy of content validity), if considering as a cut-off point that any Aiken V value > 0.70 can be considered adequate [54]. It is no less true in this respect, that out of the 68 dimensions assessed, nine did not exceed the minimum value for content validity, all with arithmetic means < 3 and with larger standard deviations (greater heterogeneity in the scores of the experts) than the remaining 59, which have arithmetic means >3 and smaller standard deviations denoting less heterogeneity in the experts’ scoring. For this reason, it is necessary to modify the items that make up these dimensions with the additional incorporation of the qualitative suggestions for improvement indicated by the experts.

In relation to the concordance or level of agreement reached among the experts, the intraclass correlation coefficient was developed (see Table 6). This coefficient is a robust technique based on repeated measures or within-subject variance analysis [55].

In this sense, the starting point is a data matrix of order *nxk*, where *n* in this case is the various aspects evaluated and *k* is the evaluators, containing each evaluation *X_ij_* of the evaluated aspect (*i*) by each evaluator (*j*). To determine the intraclass correlation coefficient, the different sources of variation (resources of variation), i.e., the various sums of squares →*SS**_suj_*: Variation between the rated aspects; *SS**_eval_*: Variation between the experts, and *SS**_res_*: Error or residual variation, must be broken down. Taking into account the results referred to in the variance analysis, as well as the formula for the development of the intraclass correlation coefficient for a two-way mixed effects model where people effects are random and measures effects are fixed, i.e.,
$$ICC=\frac{MS_{suj}-MS_{res}}{MS_{suj}+\left(k-1\right)MS_{res}+k/n(MS_{eval}-MS_{res})}$$
where *MS*_s*uj*_ represents the mean square of the rated aspects; *MS**_eva_*_l_ represents the mean square of the experts’ ratings, and *MS**_res_* represents the mean square of the error or residual.

As can be seen from the table above, in all of the cases the intraclass correlation coefficients (ICC) obtained refer to the single measures as an absolute agreement, as those referring to the average measure are in fact Cronbach’s α coefficients (consistency).

Moreover, it can be seen that the results obtained range from the smallest value for the appropriateness of the number of items CCI = 0.595 and the largest value of CCI = 0.641 referring to the relevance of the items. In any case, and taking into consideration the interpretative criteria of [55,56,57], the values obtained can be considered to be moderately agreed among the 10 experts. On the other hand, the significance values associated with each coefficient (all *p* < 0.001) reveal that the agreements, moreover, are not due to chance.

## 4. Conclusions of the Study

The management of people’s suffering due to illness, relationships, economic issues, and failure to adapt to the environment is a complex issue. It was not sufficient to define each of the categories with scientific literature, but they have also been properly quantified. In recent times, qualitative methodology has been strengthened as a procedure for obtaining scientific knowledge and limiting factors, such as the treatment of validity or the incorporation of computer programs, have been resolved.

Qualitative research is the scientific method of observation to collect non-numerical data. Instruments for measuring distress include interviews, surveys, focus groups, observation techniques, and participant observation. Qualitative research is based on case studies, personal experiences, life stories, interviews, etc. Therefore, this research does not insist on a representative sample of results. It acquires external validity through various strategies, including fieldwork and triangulation of results. Methodologically, it is an interpretative, naturalistic approach to its object of study. This means understanding reality in its natural and everyday context, trying to interpret the phenomena according to the meanings given to it by the people involved, since descriptive data are obtained such as the interviewees’ own spoken or written expressions, observable behavior, culture, and religion.

It is assumed that the information provided by qualitative techniques is just as useful and scientific as quantitative techniques. The difference lies in the type of information that each one provides. Furthermore, it should be borne in mind that there is no single form of qualitative research, but rather multiple approaches whose fundamental differences are marked by the choices made [58], and thus the use of the most appropriate techniques for collecting information.

The qualitative technique in our case is the one that has been developed, characterized by a deductive-inductive process through interviews that have brought the researcher closer to the knowledge of suffering and its management. Moreover, it provides us with a greater depth in the response and a greater understanding of the phenomenon under study. Furthermore, the interviews allow for more flexibility in their application and favor the establishment of a more direct link with the subjects.

Among the advantages provided by qualitative techniques, the following are highlighted:
-They address complex problems such as the study of the management of suffering, beliefs, motivations, and attitudes of the population. Personal suffering is very difficult to measure, in the same way as it is difficult to measure the des humanization of human beings;-They allow for the participation of individuals with diverse experiences, which provides a broader view of the problems;-A large number of ideas are generated quickly, and decision-making time is reduced.

In this piece of research, the interview was carried out, as has been seen, since it possesses suitable characteristics such as: Understanding rather than explaining, the expected answers are subjective and sincere, the interviewer listens but does not evaluate, there is maximum flexibility, as new topics arise, they are addressed, contextualized information is obtained, and there are open answers without pre-established categories, etc.

The information collected in the semi-structured interviews is of high quality, confidential, and complex. Therefore, it has an acceptable content validity for use to the extent of the criterion under consideration.

In relation to validation of the content of the questionnaire, it should be noted that, in general, a moderate relevance of each item with respect to the evaluated construct has been achieved, as can be seen from the various validity indices calculated for the different dimensions and criteria considered. In terms of agreement among the experts, moderately high intraclass correlation coefficients (ICC) were also achieved, which, more importantly, were statistically significant and, therefore, not due to chance. Notwithstanding the above, there are some considerations to be taken into account.

With regards to content validity, not all of the dimensions assessed obtained similar results. In this respect, it can be seen that the suffering dimension is that which, after the experts’ assessment, did not obtain sufficiently high validity coefficients in relation to the wording and adequate number of items. This is why it is necessary to reformulate the items that make up this dimension on the basis of the considerations suggested by the experts.

Other dimensions that also failed to achieve minimally adequate content validity indices were the dimensions love (adequate number of items), non-acceptance (relevance), spirituality (wording), gratitude (wording), sadness (relevance), resilience (content), and happiness (content). In addition to content validity, expert agreement has been considered for this study. In this sense, results were recorded that show that the degree of agreement among the experts is moderately high and, moreover, is not due to chance. Although as in the case of content validity, some dimensions achieved more agreement than others. The dimension with the greatest agreement is the relevance of the items that make up the questionnaire, while that which generated the least agreement, although sufficient, was the number of items that make up the questionnaire, which is considered as excessively high by the experts and whose findings call for reduction. For all of these reasons, and taking into consideration the suggestions for improvement made by the experts, the items of these dimensions in the aspects explained have been reformulated based on the experts’ findings, which the reader can consult as an appendix to this work.

Therefore, the experience accumulated in qualitative research cannot go unnoticed even by those who opt for epistemological positions close to the most orthodox positivism, which is to say, to that closed vision that exclusively seeks to find objectivity in what can be quantified and reduced to the merely statistical.

This article is an example of this, in which an attempt has been made to make up for the possible shortcomings of the qualitative method, using the so-called expert judgement and Aiken’s V coefficient for content validation, although improvement is always a work in progress, since expressing reality and producing knowledge is done through a dialectic process in which there is the art of persuading, debating, and reasoning different ideas in order to try to arrive at the truth.

From this point of view, relevant information is represented in order to validate the researcher’s action, which is focused on a hermeneutic rationality expressed in qualitative methods. Field information has been collected in an organized way with the construction of a priori categories, appropriate procedures have been used to analyze the information obtained from the judgement of experts, and criteria to interpret this information with the aim of providing a suitable tool to those who work in education under this perspective.

Finally, with regards to future projections, there is still a lot of work to be done. In this continuous learning process, strategies for action must be designed in context and integrated where the actors are protagonists and agents. Moreover, we understand that this must be carried out from a multidisciplinary approach that incorporates and integrates everything that has been researched in different fields. These strategies must go a step further and be implemented in the management of different areas (social, clinical, educational). It is a question of understanding in order to be able to act.

## Figures and Tables

**Table 1 ijerph-18-11393-t001:** Identification sheet of interviewees.

Identification	Sex	AGE	City of Residence	Educational Level	Profession	Religion-Beliefs
S.1	Female	41	Miami	High school	Housekeeping	Catholic believer
S.2	Male	65	Churriana de la Vega	Primary studies	Waitress	Believer
S.3	Male	50	Santa Fe	Primary studies	hostelry	Catholic believer
S.4	Female	80	Granada	Primary studies	Housekeeping	Creyente
S.5	Female	48	Granada	University education	Professor	Catholic believer
S.6	Female	44	Cúllar Vega	University education	Administrative assistant	Catholic believer
S.7	Female	40	Granada	University education	Lawyer	Catholic believer
S.8	Female	64	Miami	University education	Realtor	Catholic believer
S.9	Male	41	Granada	University education	Security guard	Catholic believer
S.10	Female	58	Granada	University education	Housewife	Catholic believer
S.11	Female	62	Granada	Primary studies	Housewife	Catholic believer
S.12	Female	50	Granada	University education	Professor	Believer
S.13	Female	47	Huétor Vega	Primary studies	Geriatric assistant	Catholic believer
S.14	Female	60	Granada	Primary studies	Housewife	Catholic believer
S.15	Male	36	Granada	High school	Businessman	Atheist
S.16	Female	46	Granada	University education	Professor	Believer
S.17	Female	57	Alhendín	University education	Technical support analyst	Catholic believer
S.18	Female	54	Granada	University education	Housewife	Catholic believer
S.19	Male	53	Granada	University education	Administrative SAE	Agnostic
S.20	Female	49	Churriana de la Vega	University education	Professor	Catholic believer
S.21	Female	42	Granada	University education	Educational counselor	Believer
S.22	Male	57	Mallorca	University education	Lawyer	Catholic believer

Source: Author’s own elaboration.

**Table 2 ijerph-18-11393-t002:** Expert’s identification data.

Expert Code	Sex	Years of Experience at the University	Department	University	Current Professional Category
1	H	25	Pedagogy	UGR	University professor (retired)
2	M	15	Business organization	UGR	University professor
3	H	40	Philanthropy and Business	Harvard Business School	Speaker and Philanthropy
4	M	20	Traduction and interpretation	UGR	University professor
5	H	17	Polítical science and Administration	UGR	Associate Professor
6	M	5	Pedagogy	UGR	Hired Postdoctoral professor
7	M	29	Pedagogy and singing	Music superior school Conservatory at Granada	Titular teacher (Special regimen)
8	M	23	Sports medicine Andalusian center (CAMD)	UGR	Sport doctor
9	M	17	Business organization	UGR	University professor
10	H	23	Pedagogy	UGR	Permanent hired doctor
Total	4 H 6 M	25 years: 1	Pedagogy: 3	UGR: 8	University professor (retired): 1
		15 years: 1			University professor: 3
		40 years: 1	Business organization: 2		Speaker and Philanthropy: 1
		20 years: 1	Traduction and interpretation: 1		Associate professor: 1
		17 years: 2	Polítical science and Administration: 1	Conservatory: 1	Titular professor: 1
		5 years: 1	Pedagogy and singing: 1		Sport doctor: 1
		29 years: 1	CAMD: 1	Harvard: 1	Permanent hired doctor: 1
		23 years: 2	Philanthropy and Business: 1		Hired Postdoctoral professor: 1

Source: Author’s own elaborations.

**Table 3 ijerph-18-11393-t003:** Validation scale.

Interview Questions				
	Nothing Appropriate	Inadequate	Fairly Adequate	Very Adequate
Clarity of content				
Clarity in drafting				
Number of questions				
Relevance of the data provided				
Proposed modification of questions:				
Questions to add:				
Questions I would delete:				

Source: Author’s own elaboration.

**Table 4 ijerph-18-11393-t004:** Categories that make up the instrument and their operational definition.

*Categories*	*Definition*
**Suffering**	Suffering is a negative emotional response, a complex and negative affective and cognitive state. It depends on the individual and the meaning given to it is subject to the fears or challenges it poses for the person experiencing it. Moreover, it depends on: Mental structure, flexibility, and adaptability. Optimal care of suffering is based on a multidimensional and continuous assessment and treatment that should be carried out in a clinical context where the psychological, physical, spiritual, and socio-cultural needs of individuals and families are simultaneously addressed [1].
**Love in the dimension of family, friends, and partner**	Love is the only way to reach the depths of a person’s personality. No one knows the essence of another human being unless he or she loves him or her. Through the spiritual act of love, the essential traits of the loved one are seen, their potential, which has not yet been revealed. Moreover, through love, the one who loves enables the beloved to realize his or her hidden possibilities. Love enables the other to realize his or her personal potential [35] (p. 139).
**Acceptance**	The moment of success comes when someone does not have to change the situation, since they have changed their thinking about the situation instead [6] (p. 326).
**Non-Acceptance**	Psychological problems are not caused by negative thoughts, sadness or anxiety, but arise when they take on a leading role and end up becoming relevant and directing the person’s choices, pushing the person’s values to the background [36] (p. 16).
**Resignation**	If I resign myself, the pain and suffering will always remain with me, I remain trapped in the situation I resign myself to, feeling sorry for myself, feeling that I am a victim of the situation and doing nothing about it, as I rarely say to myself “this is what it is, I can’t do anything about it”.I resign myself [37] (p. 2).
**Spiritual Dimension**	People live their faith (whatever that may be) with faithfulness and peace, resulting in less stress. This is due to multiple factors: Having meaning in life, being in a supportive community, purpose and goals etc., and prayer/meditation as a coping mechanism to deal with problems and difficulties all contribute to the desired inner balance. The effects will be similar in Buddhist meditation, in mindfulness, in Christian prayer, and in Jewish prayer, as long as they involve two components: Acceptance and surrender. If one asks by demanding, imploring with anguish, rather than alleviating uneasiness, one generates more unease [38] (p. 169).
**Verbal and non-verbal communication**	Discomfort sets in due to the “game of pretence” that attempts to maintain the illusion that “everything is fine”. This behavior is detrimental not only to the patient and his or her relatives, but also to the entire professional team. However, when silence is broken by dialogue and attentive and sensitive listening, everyone involved is alleviated. Efficient, respectful, and ethical communication is fundamental for all at this inevitable time [39] (p. 354).
**Pain**	An objectifiable, unitary, and tipificable sensation, most likely transmitted by specialized nerve fibers and identified by the patient as being of this type of sensation, whether pleasant or not [40] (p. 105).
**Fear**	This is a sign that indicates a discrepancy between the threats people face and the resources they have to resolve them. It is a key emotion for human survival [41] (p. 90).
**Transience**	The past is a valuable source of information, but it cannot predetermine a person’s future. Dwelling on the past, returning again and again to something that has already happened, can have harmful effects, ranging from emotions or sensations such as melancholy, frustration, guilt, sadness or resentment to depression itself.They all have one thing in common, which is that they prevent people from enjoying the present. Remaining mired in the past prevents one from moving forward in life [38] (p. 66).
**Gratitude**	This is the perception of a positive personal outcome, not necessarily deserved or earned, due to the actions of another person. It is crediting someone for positive events. Being grateful has beneficial consequences, a feeling of recognition towards others or divinity, which can be expressed in words, objects, and rituals. Whoever gives thanks expresses gratitude. It is the appreciation one has towards someone who does a favor or helps. It is a feeling that tries to return the cooperation received. Gratitude is accompanied by other feelings such as love, fidelity, and friendship. Gratitude as a value is a virtue certain individuals show to thank a person who favored them with their help [17] (p. 42).
**Compassion**	Compassion consists of five elements: Recognizing suffering, understanding the universality of human suffering, feeling for the person suffering, tolerating uncomfortable feelings, and motivation to act to alleviate suffering. It is not only about being touched by a person’s suffering, but also about wanting to act to help that person [42] (p. 15).
**Hope**	Optimism is related to hope. It consists of knowing the steps that must be taken to reach a certain goal and having the energy to do so. Hope is a motivating force, the absence of which leads to paralysis. Hope is crucial for anyone taking on hard work, and since positive expectations can be especially beneficial in the most difficult jobs, learning to be optimistic can be a very rewarding work strategy [43] (p. 184).
**Palliative care**	The aim of palliative care is to prevent and alleviate suffering, and to provide the best possible quality of life for the sick and their families, regardless of the stage of the disease or the need for further treatment. Accompanying a human being who is suffering or dying is one of the greatest challenges a caregiver can face. Caring is related to respecting the wishes of the other person, to accepting the other person as they are, to welcoming their needs, and sharing their anxieties. Caring is giving oneself in continuous presence, demanding attention and readiness for communion with the other person. The perception of their needs involves a degree of sensitivity, of reflection on values, meanings, and relationships. It demands time, internalization, openness, and the exercise of respect for others [39] (p. 353).
**Sadness**	Sadness is a negative emotion but it is also useful and evolving. It arrives when loss of any kind is felt. This emotion regulates grief, it makes people process it, it makes people take refuge within themselves until they assimilate it and their strength returns. Its function is to provide the time and the introspection required to rebuild life without what has been lost. It makes people feel compassion for themselves and repair the grief [41] (p. 50).
**Resilience**	Resilience has been defined as “a dynamic developmental process that reflects evidence of adaptation and effective coping despite significant life adversity” [44] (p. 8).
**Happiness**	Happiness is not in pain, but in overcoming pain, difficulties, and obstacles that prevent people from enjoying the authentic essence of being human [45] (p. 230).

Source: Author’s own elaboration.

**Table 5 ijerph-18-11393-t005:** Results and interpretation of the content validity calculated using Aiken’s V.

						95% Confidence Interval	
Dimensions of Scale and Criteria Considered	Number of Reviewers	Mean	Sd	Aiken V Index	Interp.	Lower Bound	Upper Bound
suf_cont	10	3.30	0.82	0.77	Valid	0.59	0.88
suf_draftr	10	2.80	0.79	0.60	Not valid	0.42	0.75
suf_number	10	2.70	0.95	0.57	Not valid	0.39	0.73
suf_relev	10	3.40	0.70	0.80	Valid	0.63	0.90
love_cont	10	3.40	0.70	0.80	Valid	0.63	0.90
love_draftr	10	3.40	0.52	0.80	Valid	0.63	0.90
love_number	10	2.90	0.88	0.63	Not valid	0.46	0.78
love_relev	10	3.20	0.92	0.73	Valid	0.56	0.86
accep_cont	10	3.30	0.67	0.77	Valid	0.59	0.88
accep_draftr	10	3.20	0.63	0.73	Valid	0.56	0.86
accep_number	10	3.30	0.67	0.77	Valid	0.59	0.88
accep_relev	10	3.40	0.52	0.80	Valid	0.63	0.90
no_accept_cont	10	3.10	0.88	0.70	Valid	0.52	0.83
no_accept_draftr	10	3.20	0.63	0.73	Valid	0.56	0.86
no_accept_number	10	3.20	0.63	0.73	Valid	0.56	0.86
no_accept_relev	10	2.90	0.88	0.63	Not valid	0.46	0.78
resig_cont	10	3.40	0.52	0.80	Valid	0.63	0.90
resig_draftr	10	3.50	0.53	0.83	Valid	0.66	0.93
resig_number	10	3.50	0.53	0.83	Valid	0.66	0.93
resig_relev	10	3.40	0.52	0.80	Valid	0.63	0.90
spirt_cont	10	3.22	0.67	0.74	Valid	0.56	0.86
spirt_draftr	10	2.90	0.74	0.63	Not valid	0.46	0.78
spirt_number	10	3.20	0.63	0.73	Valid	0.56	0.86
spirt_relev	10	3.30	0.48	0.77	Valid	0.59	0.88
verb_noverb_comm_cont	10	3.50	0.53	0.83	Valid	0.66	0.93
verb_noverb_comm_draftr	10	3.40	0.52	0.80	Valid	0.63	0.90
verb_noverb_comm_number	10	3.10	0.74	0.70	Valid	0.52	0.83
verb_noverb_comm_relev	10	3.50	0.53	0.83	Valid	0.66	0.93
pain_cont	10	3.40	0.52	0.80	Valid	0.63	0.90
pain_draftr	10	3.10	0.74	0.70	Valid	0.52	0.83
pain_number	10	3.30	0.67	0.77	Valid	0.59	0.88
pointrel	10	3.40	0.52	0.80	Valid	0.63	0.90
fear_cont	10	3.20	0.63	0.73	Valid	0.56	0.86
fear_draftr	10	3.20	0.63	0.73	Valid	0.56	0.86
fear_number	10	3.40	0.52	0.80	Valid	0.63	0.90
fear_relev	10	3.40	0.52	0.80	Valid	0.63	0.90
trans_cont	10	3.30	0.67	0.77	Valid	0.59	0.88
trans_draftr	10	3.30	0.67	0.77	Valid	0.59	0.88
trans_number	10	3.50	0.53	0.83	Valid	0.66	0.93
trans_relev	10	3.50	0.53	0.83	Valid	0.66	0.93
grat_cont	10	3.30	0.48	0.77	Valid	0.59	0.88
grat_draftr	10	3.00	0.82	0.67	Not valid	0.49	0.81
grat_number	10	3.30	0.48	0.77	Valid	0.59	0.88
grat_relev	10	3.30	0.48	0.77	Valid	0.59	0.88
comp_cont	10	3.20	0.63	0.73	Valid	0.56	0.86
comp_draftr	10	3.20	0.63	0.73	Valid	0.56	0.86
comp_number	10	3.20	0.92	0.73	Valid	0.56	0.86
comp_relev	10	3.20	0.63	0.73	Valid	0.56	0.86
hope_cont	10	3.20	0.42	0.73	Valid	0.56	0.86
hope_draftr	10	3.30	0.48	0.77	Valid	0.59	0.88
hope_number	10	3.20	0.42	0.73	Valid	0.56	0.86
hope_relev	10	3.30	0.48	0.77	Valid	0.59	0.88
palliat_care_cont	10	3.40	0.52	0.80	Valid	0.63	0.90
palliat_care_draftr	10	3.40	0.52	0.80	Valid	0.63	0.90
palliat_care_number	10	3.30	0.67	0.77	Valid	0.59	0.88
palliat_care_relev	10	3.40	0.52	0.80	Valid	0.63	0.90
sadness_cont	10	3.40	0.52	0.80	Valid	0.63	0.90
sadness_draftr	10	3.40	0.52	0.80	Valid	0.63	0.90
sadness_number	10	3.30	0.82	0.77	Valid	0.59	0.88
sadness_relev	10	2.80	0.79	0.60	Not valid	0.42	0.75
resil_cont	10	2.70	0.95	0.57	Not valid	0.39	0.73
resil_draftr	10	3.40	0.70	0.80	Valid	0.63	0.90
resil_number	10	3.40	0.70	0.80	Valid	0.63	0.90
resil_relev	10	3.40	0.52	0.80	Valid	0.63	0.90
happiness_cont	10	2.90	0.88	0.63	Not valid	0.46	0.78
happiness_draftr	10	3.20	0.92	0.73	Valid	0.56	0.86
happiness_number	10	3.30	0.67	0.77	Valid	0.59	0.88
happinees_relev	10	3.20	0.63	0.73	Valid	0.56	0.86

Source: Author’s own elaboration.

**Table 6 ijerph-18-11393-t006:** Results of the absolute agreement between experts calculated using the intraclass correlation coefficient.

Total (four dimensions)	**Measure Type**	**Intraclass Correlation ^b^**	**95% Confidence Interval**		**F Test with True Value 0**			
			**Lower Bound**	**Upper Bound**	**Value**	**df1**	**df2**	**Sig.**
	Single Measures	0.627 ^a^	0.429	0.862	115.370	8	536	0 ***
	Average Measures	0.991 ^c^	0.981	0.998	115.370	8	536	0 ***
Content of items	**Measure Type**	**Intraclass Correlation ^b^**	**95% Confidence Interval**		**F Test with True Value 0**			
			**Lower Bound**	**Upper Bound**	**Value**	**df1**	**df2**	**Sig.**
	Single Measures	0.624 ^a^	0.409	0.864	29.273	8	128	0 ***
	Average Measures	0.966 ^c^	0.922	0.991	29.273	8	128	0 ***
Drafting of items	**Measure Type**	**Intraclass Correlation ^b^**	**95% Confidence Interval**		**F Test with True Value 0**			
			**Lower Bound**	**Upper Bound**	**Value**	**df1**	**df2**	**Sig.**
	Single Measures	0.609 ^a^	0.402	0.844	27.428	9	144	0 ***
	Average Measures	0.964 ^c^	0.92	0.989	27.428	9	144	0 ***
Appropriateness of Number of items	**Measure Type**	**Intraclass Correlation ^b^**	**95% Confidence Interval**		**F Test with True Value 0**			
			**Lower Bound**	**Upper Bound**	**Value**	**df1**	**df2**	**Sig.**
	Single Measures	0.595 ^a^	0.388	0.836	25.929	9	144	0 ***
	Average Measures	0.961 ^c^	0.915	0.989	25.929	9	144	0 ***
Relevance of items	**Measure Type**	**Intraclass Correlation ^b^**	**95% Confidence Interval**		**F Test with True Value 0**			
			**Lower Bound**	**Upper Bound**	**Value**	**df1**	**df2**	**Sig.**
	Single Measures	0.641 ^a^	0.437	0.860	31.290	9	144	0 ***
	Average Measures	0.968 ^c^	0.93	0.991	31.290	9	144	0 ***

Two-way mixed effects model where people effects are random and measure effects are fixed. ^a^ The estimator is the same whether the interaction effects is present or not. ^b^ Type a intraclass correlation coefficients using an absolute agreement definition. ^c^ This estimate is computed assuming that the interaction effect is absent, since it is not estimable otherwise. *** *p* < 0.001. Source: Author’s own elaboration.

## Data Availability

The datasets generated and / or analyzed during the current study will not be publicly available due to privacy and confidentiality reasons, but they will be available from the corresponding author upon reasonable request.

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
