# Peer review of "Content Validation of a Semi-Structured Interview to Analyze the Management of Suffering"

_ijerph, 2021, doi:10.3390/ijerph182111393_

Round 1
Reviewer 1 Report
Dear authors
The article clearly presents the advantages that assessing people's suffering can offer to working holistically with them in some areas. However, it would be convenient to mention that the instrument presented was aimed at the population.
It would also be advisable to cite other instruments that have been used previously to measure suffering.
Another aspect that I recommend clarifying is how the 153 items and 17 categories that are statistically evaluated have been obtained. In the conclusions section it can be clarified that it was through a review of the scientific literature. A clarification in this regard would be advisable.
In the conclusions, the advantages of qualitative versus quantitative research are exposed, and it can be determined that the presented instrument has advantages over other instruments, it would be advisable to show other instruments that measure suffering and what benefits and disadvantages the presented instrument presents. with respect to others
Greetings
Reviewer 2 Report
This research has an original objective and a meaningful content. I congratulate the authors for the work done. I am grateful with the editors for the possibility of revising this manuscript. Although the quality of the manuscript is high, I would like to make some contributions that I hope will increase it and improve readers' understanding.
Introduction:
The introduction is clear and well worked.
Material and methods:
The study of the design is appropriate and well described, although the selection of the panel of experts may generate certain doubts: what are the inclusion and exclusion criteria of the same? An attached table showing these criteria could clarify the doubts.
The quality of resolution of the formulas provided (lines 215 and 225) should be improved, putting them in the same format as the formula on line 275.
Data analysis:
The statistical analysis is correct and well described.
Discussion:
it is well oriented.
Reviewer 3 Report
The paper entails the content validation of a semi-structured interview for the analysis of the management of suffering. This subject is interesting since suffering is a subjective factor related with the several dimensions of the human life
I have several points
Introduction
It should be more structured:
- Definition of human suffering
- Description of the main applied fields were analysing suffering should be usefull.
- The statement of the need to have a semi-structured interview for manage suffering.
In this part, the methodology aspects of the validation should be avoided. It should be State in the following part (Methods)
Methods
It should be exposed the method of analysis with the detail the statistical analysis and the Package used to carry out.
Discussion
It miss a paragraph about discussion of the results according to the literatura rewieded. Part of this discussion is stated now in the conclusions, mainly for the validation content. Moreover, it should be stated also the future direction of the research in this field, and in the autors study.
Conclusions
In this part, it is necessary make a conclusion about the results, the semi-structured interview could be used? Or not? And why yes or not? And in which applied fields could be implement.
Author Response
see in the attachment

Round 2
Reviewer 1 Report
Dear author,
Thank you for your answers
Author Response
thanks
Reviewer 3 Report
The manuscript has improved significantly, however I still have concerns about the methodology section and the conclusions.
In the methodology the content of each section is not clear, in some cases there is information that must be in another section.
Advice that will be structured according to standard sections and the pertinent information will be incorporated in each of them: Participants (People, form of inclusion, ethics); Content validation (include experts) and the entire validation procedure (experts, validation criteria, ..); Instrument; Statistical analysis (all the statistics that have been used (Alikens V; ICC and program used with its version).
Regarding the conclusions, it is not clear to me if the instrument is valid to be used with the reformulation of experts and in what contexts it could be applied (clinical, social, ...).
Author Response
REPLIES
Instrument: 168-211
The instrument used in this research was the semi-structured interview specifically designed on an ad hoc basis and validated by expert judgement. Based on the categories that emerged as a result of the literature review, the objectives of this research will be answered. This interview has been called "the management of suffering in people suffering from illness, relationships, economic problems and adaptation to the environment" (see appendix 1).
The sequence of items is divided into the following categories of analysis:
Socio-demographic data
Suffering
Love in the dimension of family, friends, partner
Acceptance
Non-acceptance
Resignation
Spiritual dimension
Verbal and non-verbal communication
Pain
Fear
Transience
Gratitude
Compassion
Hope
Palliative care
Sadness
Resilience
Happiness/life satisfaction, well-being
Through each of these categories, a series of items have been elaborated in order to obtain the information addressed.
The data collection took place in different places as each informant preferred, although always in quiet places that safeguarded confidentiality. The process was recorded on audio and later transcribed
Finally, state that each of the interviews carried out is developed in a different way, taking into account the context and the reporting subject. Said interviews have been recorded on audio with the consent of each of the interviewed subjects in order to preserve the collection of informative data.
Therefore, this research is framed within the proposed ethical considerations: informed consent, avoiding deception of research participants, respecting participants' privacy, upholding accuracy of data and interpretation, and respect for the individual. We also state that, with respect to the confidentiality of the participants, any identifying data that could recognize them has been removed, thus preserving anonymity in each and every one of the interviews, and with respect to the consent document, each and every one of them was informed of the purpose of the research and agreed to it.
The semi-structured interviews provided an abundance of data, which was refined and analyzed to arrive at a final result. This research study was marked by data saturation.
Participants: 213-246
It should be mentioned that the questionnaire is aimed at a sample of 22 respondents. With regard to the section on suffering due to illness, it was carried out with chronically ill patients from hospitals in Granada (Spain). On the part of the hospital, in a direct and intentional selection sample, those subjects with the highest degree of loneliness were assigned to the research, so the type of sampling was incidental. The participating sample consisted of four subjects of different genders between 41 and 80 years of age, from different cities in Spain and America.
With regard to the other sections on suffering due to economic problems, relationships and adaptation to the environment, the participant sample was made up of informants of different genders between 40 and 65 years of age (table 1, identification sheet) who provided data explaining the phenomenon of suffering until the information was saturated. Therefore, the sampling is non-probabilistic and intentional. As [32] points out, the final number of the sample is obtained when the informants do not present any more answers to the explanation of the phenomenon, reaching saturation of the testimonies or information.
Similarly, the so-called "snowball" technique was also used in sampling. This is a non-probability sampling technique used by researchers to identify potential subjects in studies where subjects are difficult to find. What is relevant in this research work is the importance of the word of the people interviewed, the information provided by them, because it is thanks to them that the results of this work will be obtained. These ideas have been defended by [33] who supports the study of narrative as it is the way in which human beings experience the world. This general notion carries over to the conception that education is the construction and reconstruction of personal and social stories; teachers and learners are narrators and characters of their own and others' stories. This concept can be applied in the interaction between patient/educator, interviewee/educator, learner/educator as mutual learning takes place through the knowledge of the informants' life stories. Thanks to these life stories, it will be possible to intervene in how to teach how to manage, lessen or alleviate suffering. As [34] explained, healing is bilateral, there is mutual teaching and learning on both sides.
Content validation: 268-272
All of them are people with extensive knowledge and proven experience in the area of interest and are therefore qualified to answer the questions posed. They come from different educational backgrounds, approach the problem experimentally as opposed to theoretically and from different professional experiences. They are intended to help improve the quality of the analysis.
Conclusions of the study: 553-569
Therefore, the experience accumulated in qualitative research cannot go unnoticed even by those who opt for epistemological positions close to the most orthodox positivism, that is to say, to that closed vision that exclusively seeks to find objectivity in what can be quantified and reduced to the merely statistical.
This article is an example of this, in which an attempt has been made to make up for the possible shortcomings of the qualitative method, using the so-called expert judgement and Aiken's V coefficient for content validation, although improvement is always a work in progress, since expressing reality and producing knowledge is done through a dialectic process in which there is the art of persuading, debating and reasoning different ideas in order to try to arrive at the truth.
From this point of view, relevant information is represented in order to validate the researcher's action, which is focused on a hermeneutic rationality expressed in qualitative methods. Field information has been collected in an organised way with the construction of a priori categories, appropriate procedures have been used to analyse the information obtained from the judgement of experts and criteria to interpret such information with the aim of providing a suitable tool to those who work in education under this perspective.